# Axial Crystal Growth Evolution and Crystallization Characteristics of Bi-Continuous Polyamide 66 Membranes Prepared via the Cold Non-Solvent-Induced Phase Separation Technique

**DOI:** 10.3390/polym14091706

**Published:** 2022-04-22

**Authors:** Jiangyi Yan, Lihong Nie, Guiliang Li, Yuanlu Zhu, Ming Gao, Ruili Wu, Beifu Wang

**Affiliations:** 1College of Petrochemical Engineering and Environment, Zhejiang Ocean University, Zhoushan 316000, China; jiang1842022@163.com (J.Y.); nielihong1975@163.com (L.N.); ligl@zjou.edu.cn (G.L.); xiaoluer113@163.com (Y.Z.); gaoming97@163.com (M.G.); 2Sichuan Bureau of National Food and Strategic Reserves Administration, Chongqing 401326, China; wuruili@163.com

**Keywords:** polyamide, membrane, crosslinked structure, microporous structure

## Abstract

Polyamide 66 microporous membranes were prepared by cold non-solvent-induced phase separation using polyamide 66-formic acid-propylene carbonate as a ternary membrane-forming system. The formed membranes exhibited a special bicontinuous structure consisting of interglued spherical crystals or interlocked bundles of microcrystalline aggregates. The variation of the microporous structure under the influence of preparation conditions, solvent, aging time, and polymer concentration affects the comprehensive performance of the membranes. For example, the cold-induced operation and the use of different membrane-forming solvents contributed to the crystallization of polyamide 66, resulting in an increased contact angle of polyamide 66 membranes, obtaining a high resistance to contamination of up to 73.5%. Moreover, the formed membranes still have high mechanical strength.

## 1. Introduction

Polyamide, particularly polyamide 6 or polyamide 66, is currently the most produced and widely used variety of polyamide plastic. Polyamide 66, also known as nylon 66, is a form of opalescent crystalline polymer with excellent elasticity, strength, chemical resistance, high melting point, high elongation, easy processing, and low production costs, and it is widely used in various fields. Polyamide 66 is a type of crystalline linear macromolecule with relatively high regularity in chemical structure, with no branched chains on the molecular chain, and the presence of amide-based polar groups makes the arrangement of macromolecular chain stacking regular, generally in an extended-plane serrated structure. In polyamide 66, the long polymer chains are staggered, allowing hydrogen bonds to form even without molecular deformation, which makes crystallization more stable and connected into sheets. It is because of this unique molecular structure that polyamide 66 polymer can be modified to make its performance excellent [1,2,3].

Polyamide 66 has long been used to produce microporous membranes for fine separation processes. These membranes have a rough, porous surface structure with a symmetrical bicontinuous interior that provides high permeate fluxes in filtration operations, not only because they are water-wettable but also because the feed fluid is driven by the microcapillary forces of the micropores as they permeate through the membrane [4,5].

Microporous polyamide membranes are generally prepared by the non-solvent induced phase separation method [6,7], which is insoluble in common organic solvents due to the strong chemical stability of polyamide. Due to the influence of polymer concentration, temperature, additives, and other conditions in the casting membrane system, the formed membranes will exhibit symmetric or asymmetric structures [1,8]. 

In order to develop membranes with continuous structure, an optimized membrane preparation method was adopted in this study to combine some advantages of thermally induced phase separation [9,10,11]. The developed polyamide 66 membrane formed an internal continuous structure system with the coexistence of microcrystals and pore networks and was characterized and analyzed by relevant technical means.

The preparation, structural characterization, and performance of the new polyamide 66 membranes are detailed in the sections that follow.

## 2. Experimental

### 2.1. Materials

Polyamide 66 (PA66, A.R) was provided by French manufacturer Rhodia Co., Ltd., Bangkok, Thailand; Formic acid (HCOOH, FA, 94%) was purchased from Yangzi Petrochemical-Basf Co., Ltd., Nanjing, China; 1,2-propanediol carbonate (C_4_H_6_O_3_, PC, A.R) was purchased from National Pharmaceutical Chemicals Co., Ltd., Beijing, China; and bovine serum albumin (BSA, Mn = 66,000) was purchased from West Asia Chemical Technology Co., Ltd., Jinan, China.

### 2.2. Preparation of PA66 Flat Sheet Membranes

Polyamide 66 membranes were prepared in the form of flat sheets by the cold-non-solvent induced phase separation (CIPS) method. An appropriate amount of nylon 66 was mixed with FA and PC in a glass bottle sealed with a Teflon-lined cap. The mixture was blended on a magnetic stirrer at 60 °C until a clear homogeneous solution was formed. The solution was aged for a period of 2 h in a thermostat held at 45 °C to remove air bubbles. The solution was naturally cooled to room temperature. Then, the casting solution was coated on the glass plate with a self-made scraping glass rod. The cast glass plate was left for 3 min at ambient temperature before being immersed into a water coagulation bath at 15 °C for the phase inversion process to take place. Once the membrane was peeled off from the glass plate, it was transferred to another water bath and kept for 24 h to ensure the complete removal of residual FA. Before usage, the membranes were washed by deionized water for three times. The cleaned membrane was dried for subsequent characterization. The composition and preparation conditions of different membrane samples were shown in Table 1.

### 2.3. Membrane Characterizations

#### 2.3.1. Basic Physicochemical Characterizations

X-ray photoelectron spectroscopy (XPS Thermo Scientific K-Alpha, Waltham, MA, USA) was used to investigate the surface chemical compositions of membranes. The range of survey spectra is from 0 to 1300 eV, and the C1s’ peak of high-resolution spectra was detected. XPS full-scan spectra were recorded within the range from 0 to 1300 eV with a pass energy of 150 eV with a monochromatic Al Kα X-ray source at 1486.6 eV. Fourier transform infrared spectroscopy-attenuated total reflectance (FTIR-ATR) measurement was carried out using a Thermo Scientific Nicolet iS20 Fourier transform infrared spectrometer. The samples were placed on the sample holder, and all spectra were recorded from 4000 cm^−1^ to 600 cm^−1^. Nuclear magnetic resonance (NMR) analysis was performed on the developed membrane to further determine its chemical structure by using C6D6 as a solvent. Proton nuclear magnetic resonance spectrum (^1^H NMR) was recorded on a Bruker (Berlin, Germany) 400 MHz spectrometer. Chemical shifts (δ) are reported in parts per million (ppm). The relaxation delay time is 3 s.

#### 2.3.2. Crystalline Forms

The crystal structure of polyamide 66 in the prepared membranes was determined by means of wide-angle X-ray diffractometer (WAXD, DX-2700, Shanghai, China). The scanning parameters included the source intensity (40 kV/30 mA), λ (1.54 Å, Cu Kα line), source slit width (0.6 mm), increment rate (1.20°/min), and scanning range (5–80°). The crystal size of PA66 in the membranes was estimated by Scherrer’s equation:(1)$$D=\frac{K\lambda }{\beta COS\theta }$$
where D is the estimated diameter of the crystals (nm); K is the Scherrer’s constant (K = 0.89); λ is the wavelength of the incident X-rays (nm), which is 0.154 in this study; β is the peak width at half height (rad); and θ is the diffraction angle (rad).

The melting and crystalline properties of PA66 membranes were studied using a differential scanning calorimeter (DSC 200-F3, NETZSCH, Germany) in a nitrogen atmosphere. About 8 mg of dry membrane was sealed in an aluminum pan. It was kept at 20 °C for 3 min, then heated to 400 °C at a heating rate of 10 °C/min and maintained 3 min to remove the thermal history, and then chilled to 20 °C (the rate of 10 °C/min), during which the melting curves were recorded. All tests were carried out under dry nitrogen atmosphere at a pressure of 0.05 MPa, and the flow rates of purge gas and protective gas were 80 mL/min and 30 mL/min, respectively.

#### 2.3.3. Membrane Morphology and Microstructure

The morphology of the cross section and surface for various membranes was observed by scanning electron microscope (SEM Gemini 300, ZEISS, Jena, Germany) with an accelerating voltage of 15 kV. The upper surfaces and cross-sectional morphologies of various membranes were examined using a scanning electron microscopy (SEM Gemini 300, ZEISS, Jena, Germany) with an accelerating voltage of 15 kV. When characterizing cross-sectional morphologies, the samples were prepared by fracturing membranes after fully cooling in liquid nitrogen. All samples were treated with Au/Pd sputtering and carefully handled to avoid contamination. 

#### 2.3.4. The Hydrophilicity of Membrane

The hydrophilicity of the membrane was observed based on the water contact angle measurement (WCA). The WCA of the membrane was measured using an optical contact angle tester (OCA15Pro, DataPhysics Instruments GmbH, Filderstadt, Germany) according to the sessile-drop method. This contact angle can be described as an angle between the sample surface and calculated drop shape function, the projection at which the drop image is referred to as the baseline. Briefly, a water droplet was deposited on the membrane surface, and the value of contact angle was observed and recorded until there was no change of droplet during the short measurement period. The average was made by measuring parallel three times for each sample. 

#### 2.3.5. The Permeability, Anti-Fouling Performance of the Membrane

Pure water flux (PWF) was measured with a cup ultrafilter at 0.15 MPa pressure. The flux was equilibrated for the passage of the first 30 min of permeation, whilst the following 10 min of permeation was collected. Pure water flux was evaluated by the following:(2)$$J=\frac{V}{A\cdot \Delta t}$$
where J is pure water flux (L/(m^2^·h)), V is the volume of penetrated water (L), A is the effective area of the membrane (m^2^), and Δt is the recorded time (h). All experiments have been conducted thrice to obtain the results presented in this study, which were an average value. 

UV/visible spectroscopy (TU-1901, General analysis) was employed to measure the concentration of feed solution and permeation of the BSA solution at a wave-length of 280 nm. The feed solution was 0.1 g/L BSA solution. Then, the filtered solution was obtained in the same way as water flux testing. The membrane rejection (R) was obtained by Equation:(3)$$R=\frac{C_{F}-C_{P}}{C_{F}}\times 100\%$$
where R is the percentage retention rate or protein (BSA) rejection, $C_{F}$ is the concentration of feed solution, and $C_{P}$ is concentration of permeation. 

The antifouling performance of various pure PA66 membranes was investigated using BSA. This protein was selected as the model foulant because it is one of the common membrane foulants. The foulant (Feed) solution was also 0.1 g/L BSA solution. The protocol for fouling tests involved four main stages, which included the recording of the water flux at 0.15 MPa, subjecting the membrane to a feed solution for one hour and backwashing with pure water for one hour, and the flux recovery rate (FRR) was calculated by the following equation:(4)$$FRR=\frac{J_{W2}}{J_{W1}}$$
where $J_{W2}$ is the PWF after washing of the membrane, and $J_{W1}$ is the initial PWF before fouling the membrane with BSA. 

#### 2.3.6. The Porosity of the Membrane

After measuring the dry weight of the membranes, they were immersed in n-butanol for 12 h to become wet, and the wet weight was then measured. The porosity for the membranes was obtained by the following equation:(5)$$\varepsilon =1-\frac{W_{1}/\rho_{m}}{(W_{0}-W_{1})/\rho_{d}+W_{1}/\rho_{m}}\times 100\%$$
where ε is porosity (%), $W_{1}$ is the mass for the dry membrane (g), and $W_{0}$ is the mass of membrane (g) after absorbing n-butanol. $\rho_{d}$ and $\rho_{m}$ represented the density of n-butanol (g cm^−3^) and membrane (g cm^−3^), respectively.

#### 2.3.7. Mechanical Property of Membrane

The microcomputer-controlled electronic universal testing machine (CMT8501, Shenzhen, China) was employed to determine the breaking strength and breaking elongation for the membrane by measuring the stress–strain curve. The size of the selected membrane was 70 mm × 10 mm, and the distance between the two clamps was 30 mm. Each sample was tested three times to obtain average strength. They were obtained by Equations (6) and (7), respectively.
(6)$$R=\frac{F}{S}$$
where R, F, and S are the breaking strength (MPa), the breaking force (N), and the area of cross section for the membrane (mm^2^), respectively.
(7)$$\varepsilon =\frac{L}{L_{0}}$$
where ε, L, and $L_{0}$ represented break elongation (%), final length (mm), and initial length (mm), respectively.

## 3. Results and Discussion

### 3.1. Chemical Characterizations

To explore chemical compositions of membrane surfaces, XPS has been performed [12].

The XPS full-scan spectrum (Figure 1a) showed that PA66 membranes formed by different solvents contain oxygen (531 eV), nitrogen (399 eV), carbon (285 eV), and their atomic contents were listed in Table 2, respectively. Oxygen, nitrogen, and carbon were the inherent elements of polyamide 66 [13]. After the addition of PC solvent, as the oxygen in the amide carbonyl group is the only oxygen species in the PA66 material, the reduction in the atomic ratio O/C indicates that the ratio of oxygen-containing functional groups on the surface of the membrane decreases. This may be due to the different degrees of flexibility of different solvents on the polymer molecular chain, the binding effect between chain segments is different, which changes the distribution and concentration of polar groups near the membrane surface.

From the spectral graph of C1s, for different PA66 membranes (Figure 1b,c), the spectra can be deconvoluted into three peaks located at 284.8 eV, 285.9 eV and 287.8 eV attributing to C–C, N–C (amide nitrogen) and N–C=O groups [14]. Interestingly, the deconvolution O1s spectrum of the membranes consists of two peaks (Figure 1d,e). Of the two peaks, the main peak at 531.2 eV is attributed to the carbon–oxygen double bond (C=O) in the amide carbonyl group, while the second peak observed at 532.6 eV is caused by the combination of oxygen and carbon atoms alone. This is due to the presence of low level of photooxidation center carbons on the surface of the commercial PA66 material purchased [15,16]. However, the O–C=O group (288.5 eV), which is unique to propylene carbonate, is not shown in Figure 1c, indicating that the ring structure of propylene carbonate is opened, which may be caused by formic acid. 

In addition, Fourier transform infrared spectroscopy (FTIR) was used to analyze chemical composition changes on the surface of the developed PA66 membranes (Figure 2) [12]. It can be clearly seen that PA66 membranes contain a large number of related functional groups. Both spectra showed the existence of N-H stretching vibration (3295.14 cm^−1^), N-H shear vibration and C–N stretching vibration (1533.00 cm^−1^); C=O stretching vibration (1629.19 cm^−1^); and C–N stretching vibration (1415.93 cm^−1^). The characteristic peaks at 2932.48 cm^−1^ and 2858.71 cm^−1^ all correspond to the stretching vibration of methylene (–CH_2_). Furthermore, small peaks at 1310–1200 cm^−^^1^ correspond to C-N stretching vibration and N-H shear vibration in polyamide [17]. When different solvents were used for membrane formation, the infrared peaks of 906 cm^−1^ and 934 cm^−1^ representing the crystallization region remained unchanged, indicating that the crystal shape of PA66 remained unchanged during membrane formation. This stems from the fact that there is only one arrangement of hydrogen bonds in polyamide 66, for which its hydrogen bonds are directly arranged in a row, which can form a strong and dense polymer structure and make it have a more stable crystallization, as shown in Figure 3. It is noteworthy that the carbonyl absorption peak at 1790 cm^−1^ of propylene carbonate in the P3 membrane disappeared when using PC solvent for membrane formation. Instead, the absorption peak of the ether bond appeared at 1110.2 cm^−1^, which did not exist in the P1 membrane. It shows that the presence of formic acid caused the ring-opening polymerization reaction of propylene carbonate, and a partial decarboxylation reaction occurred during the polymerization process to generate a partial polyether structure. These results were consistent with those obtained by XPS and affected the performance of the membrane. 

Figure 4 represents the ^1^H NMR spectrum of the PA66 membrane. The signals show chemical shifts (ppm) at 1.25 and 3.58, which indicate protons from –CH_3_ (methyl) and –CH_2_ (methylene), respectively, while the signal at 5.21 ppm represents protons from –CH_2_COOH (carboxymethyl) [18]. In formic acid, ring-opening polymerization of propylene carbonate takes place, which is consistent with the results obtained by XPS and FTIR.

### 3.2. Crystals of the Membranes

In order to further obtain information related to the polymorphisms of PA66 membranes in this study, XRD measurements were carried out [19]. Figure 5a shows the diffraction patterns of various PA66 membranes. There are two diffraction peaks at 2θ values of about 20.1 and 23.9, which correspond to crystal types α1 (100) and α2 (010/110) of PA66, respectively, proving that crystal type α is formed during the membrane-forming process. It is worth noting that the distance between the two diffraction peaks of the P0 membrane is smaller than that of other kinds of membranes, and the diffraction peak of the α2 crystal of PA66 changes obviously, and the peak angle value weakens (around 23.74), which is related to the structure of the α2 crystal. This is due to the low concentration of P0 casting solutions, resulting in loose membrane structure, while the α2 crystal in PA66 is a kind of sparse accumulation structure, which makes it easier to occupy a large number of α2 crystals in the formed membrane, which is manifested by an increase in distance between α2 crystals. Therefore, it is reflected in the XRD curve that the angle value of the diffraction peak of α2 crystalline type is weakened. In addition, according to Scherrer’s formula, the crystal size of α2 decreases in the P0 membrane, leading to a decrease in crystallinity of PA66 in the P0 membrane [20], which is consistent with the analysis result obtained by DSC and confirmed by subsequent SEM images. 

DSC curves of different kinds of PA66 membranes prepared are shown in Figure 5b. There were no significant differences in the tendencies among membranes prepared with different polymer concentrations because the solution temperatures and quenching temperatures among different membranes were the same. However, there are still some different changes that are worth studying and analyzing. A wide endothermic peak (100–200 °C) and a double endothermic peak (250–266 °C) were observed in all curves, which were consistent with the results reported in the literature for PA66, and the appearance of the endothermic peak was related to the Brill transformation of the membrane sample during the heating process. Since the formation mode of hydrogen bonds between molecular chains in PA66 is the main factor determining the crystal structure of PA66, Brill transformation serves as the equilibrium point of crystal structure transformation, so the change of hydrogen bonding surface is the causative factor for the occurrence of Brill transformation.

Compared with the sample of P0P1P6 membrane, the fluctuation of the endothermic curve of the P2P3P4P5 membrane is larger, and the peak value is closer to 200 °C with the increase in PC content, indicating that PC acts as a “lubricant” in the crystallization process of the polymer and promotes the slip between sheet crystals within the crystalline phase of the polyamide. In the nitrogen atmosphere, the curve was basically smooth at 300 °C, indicating that PA66 maintained short-term stability. After that, the curve fluctuated and PA66 decomposed to produce ammonia and carbon dioxide, etc. Among them, the fluctuation temperature of the P4P5 membrane is more advanced and changes around 315 °C, and the P5 membrane occurs earlier than P4, which indicates that the addition of PC in the casting solution changes the cross-linking degree between PA66 macromolecular chains to a certain extent, leading to the main chain being prone to cracking and thermal decomposition [15,19].

### 3.3. Morphological Characterizations

The cross-sectional image of the membrane was shown in Figure 6. It can be clearly seen from Figure 6 that there are significant differences in the morphological structure of PA66 membranes after using different solvents. A dense packing structure with interlacing and fusion of spherulites is formed for the P0P1P6 membrane (Figure 6a,b,g). With the increase in polymer concentration, the density of spherical crystals increased, and the degree of interweaving between them increased. Interestingly, many of the spherical structures are covered with micropores due to the temperature difference effect between the polymer solution and the coagulating bath so that not only mass transfer but also heat transfer occurs during the process of phase transformation. This degree of supercooling acts as the driving force of PA66 crystallization, making the spherical structure of the spheres become increasingly perfect, and the internal particles become increasingly enriched. This further densifies the PA66 membrane. This was confirmed by subsequent membrane permeability performance tests.

In contrast, for P2P3P4P5 membranes (Figure 6c–f), they showed the axial growth of petal sheet-like axial crystals with interlocking aggregates at the front and open ends. Their crystal shape indicates that they are in an early stage of spherulite maturation. The crystal evolution process is shown in Figure 7. Among them, the axial crystal shape of the P3 membrane is the most complete, which indicates that a small amount of PC solvent contributes to the crystallization of PA66. Later, with the addition of large amounts of PC, the end sheets become smaller and the end fans become close together, showing large and robust axial crystal columns, which are interlocked similarly to the trunk of a tree. In addition, it can also be found that when the amount of PC is the largest, there are some axial defective pores in the membrane, which indicates that the P5 membrane has a looser support structure compared with other membranes [21]. The different morphologic characteristics of the membranes formed by different casting solvents are not only affected by cooling temperature but also due to the fact that in the process of solvent and non-solvent exchange, they are not simultaneously displaced outward, and part of the solvent diffuses inward due to the superior solubility of formic acid to water compared to propylene carbonate [4,5]. To a large extent, the different crystallization characteristics can affect the physical, chemical and mechanical properties of PA66 membranes.

The upper surfaces of various PA66 membranes were shown in Figure 8. The morphology of the PA66 membrane is significantly affected by the polymer concentration and the induced temperature as the polymer solution can be brought into different separation regions in the phase transition by changing the composition and temperature [22]. The P0 membrane (Figure 8a) has obvious defective macropores, as seen in the magnified view of the upper surface. As polymer concentration increases, the cross-linking degree of the membrane surface increases, the entanglements of the polymer molecular chains increase, and the membrane pore size decreases, as shown in Figure 8b,g (P1P6 membrane). However, the surface porosity of P2P3P4P5 membranes with fan-shaped aggregates is affected by the amount of PC [5]. Compared with the P2 membrane, although the P3 membrane fan-shaped becomes smaller, an offset stacking occurs between the upper and lower layers, avoiding the formation of continuous pore channels.

On the contrary, in the presence of a large number of PC, there are many micropores on the fan-shaped wafers of P4P5 membranes, resulting in a decrease in the density of the membrane surface and an increase in the porosity of the membrane surface. This indicates that a large number of PC acts as a pore-forming agent to some extent.

### 3.4. Contact Angle Characterization

In order to explore the hydrophilic and hydrophobic properties of PA66 membranes, the contact angles of each membrane developed in this study are shown in Figure 9. The hydrophobic properties of the surface of the polyamide 66 material allow all membranes to have a large contact angle, which confirms Casey’s law: The roughening of a hydrophobic surface increases its apparent water contact angle [23]. The decreasing trend shown in the figure is due to the high porosity on the surface of the P4P5 membrane, as shown in the SEM figure, which expands the distribution range of pore size and enhances the wettability of the membrane’s surface to water.

### 3.5. Membrane Filtration Performance

To evaluate the separation and retention performance of the membranes, BSA filtration experiments were performed. The results of permeability flux, BSA rejection, and porosity are shown in Figure 10 and Figure 11. The permeability flux firstly decreased, then increased and then decreased, while the trend of the rejection rate was the opposite. Both P3 and P6 membranes had low flux values of 16.3 L/m^2^·h and 8.5 L/m^2^·h, respectively, but rejection rate increased by 1.36 times. At the same cooling temperature, this is attributed to the increase in polymer concentration, which provides the number of molecules for an increase in the number of crystals and the densification of the membrane [24]. 

As can be seen from the porosity diagram of the membrane, the overall trend of porosity remains at about 53.7%, but the P6 membrane has a lower porosity, which is because the increased viscosity of the casting solution makes the vacancies and pores in the membrane filled with crystal nuclei, making it more difficult for pores to form and grow. However, P4P5 membranes have higher porosity, which is due to the dilution effect caused by a large amount of PC at the membrane–bath interface, which is consistent with the results obtained from SEM image observation and analysis [16]. 

### 3.6. Anti-Fouling Performance of Developed Membranes

The antifouling performance of various PA66 membranes was studied, and the flux recovery rate was taken as an important parameter to examine the water flux recovery of the membranes after protein permeation. According to Figure 12, the flux recovery rate (FRR) of the P0 membrane was only 21.1%. The reason is that the larger pores of the P0 membrane can easily accumulate foulants without the strong shear force between water and accumulating proteins when water permeates. Interestingly, the FRR values of the membranes increased significantly when the polymer concentration increased, with P1 membranes increasing about 2.9 times more than P0 membranes, and the FRR values of the membranes developed with the addition of PC solvent were all better than those developed with a single solvent. This is because the tendency of the former membrane to contaminate is a slower process, and the latter is more rapid due to the influence of different micropore structures.

The amide group in PA66 contains carbonyl group, and it is easy for water molecules to form hydrogen bonds so as to enhance the passing ability of water molecules and to weaken the hydrophobic effect between pollutants and the membrane, which helps to reduce the pollution degree of the membrane. However, a large number of pollutants are easily embedded in the internal cross-linked structure of PA66 membranes, which leads to the adhesion of pollutants on the membrane’s surface and the formation of pollution layer at the initial stage of membrane filtration, and even the formation of membrane pore blockage [22,25]. Although the pollutants covering the membrane surface are basically removed after backwashing, there are still a large number of pollutants in the membrane pore, and the blockage of the membrane pore after physical cleaning is irreversible.

### 3.7. Mechanical Performance of the Membrane

As expected, PA66 membranes maintained high mechanical strength. This is due to the good mechanical properties of PA66 material itself [20]. Because PA66 contains a large number of amide groups and can be crystallized, there are hydrogen bonds between molecules, which enhance the intermolecular force and provide PA66 with characteristics of high strength. From Figure 13, although the elongation at break of all membranes was almost the same (about 101%), the tensile strength of the membranes developed using a mixture of FA and PC was superior to that made by a single solvent. This may be because the addition of PC in the polymer solution produces a PA66 polymer chain that is more flexible, the original fixed regularity of the PA66 molecular chain is changed, and elastic properties increased.

## 4. Conclusions

As a kind of semi-crystalline polymer, polyamide 66 membranes formed by the CIPS method exhibit a special discontinuous structure composed of interlocking crystals crosslinked with each other. The strong hydrogen bonding between PA66 molecules gives it good crystallinity, stability and excellent mechanical properties. It can be found that the formed P3 membrane achieved an optimal overall performance, with 46.4% interception capacity and 67.3% pollution resistance.

## Figures and Tables

**Figure 1 polymers-14-01706-f001:**
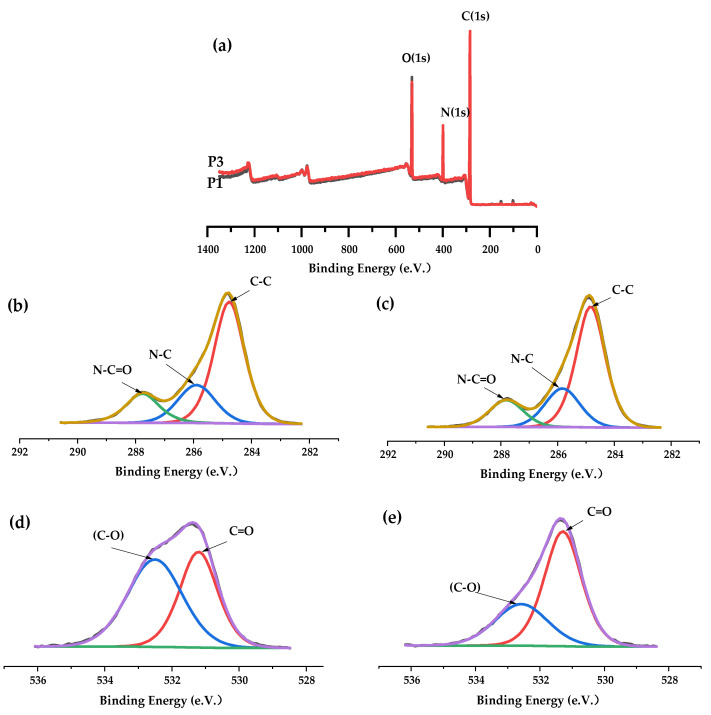
(**a**): XPS full-scan spectra of P1 and P3 membrane; (**b**,**c**): C1s spectra of P1 and P3 membrane; (**d**,**e**): O1s spectra of P1 and P3 membrane.

**Figure 2 polymers-14-01706-f002:**
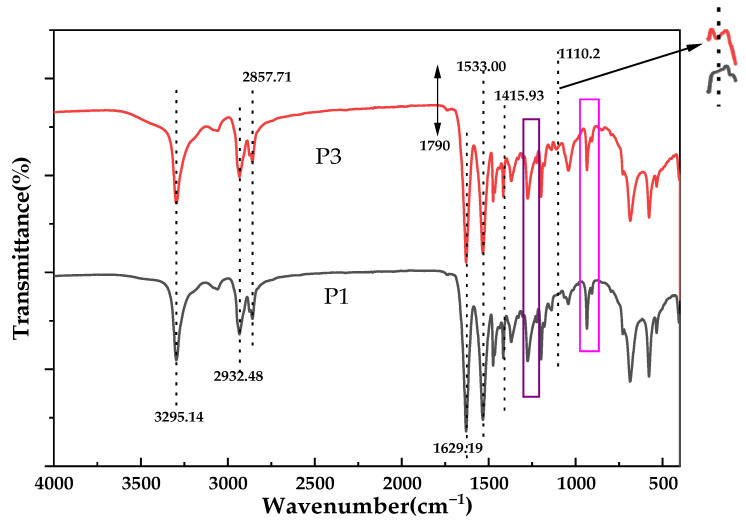
FTIR spectra of different kinds of PA66 membranes.

**Figure 3 polymers-14-01706-f003:**
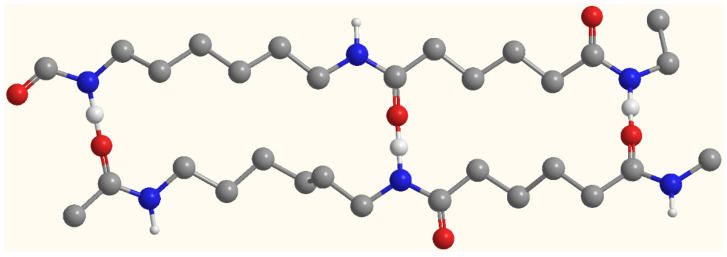
Hydrogen bond arrangement of polyamide 66.

**Figure 4 polymers-14-01706-f004:**
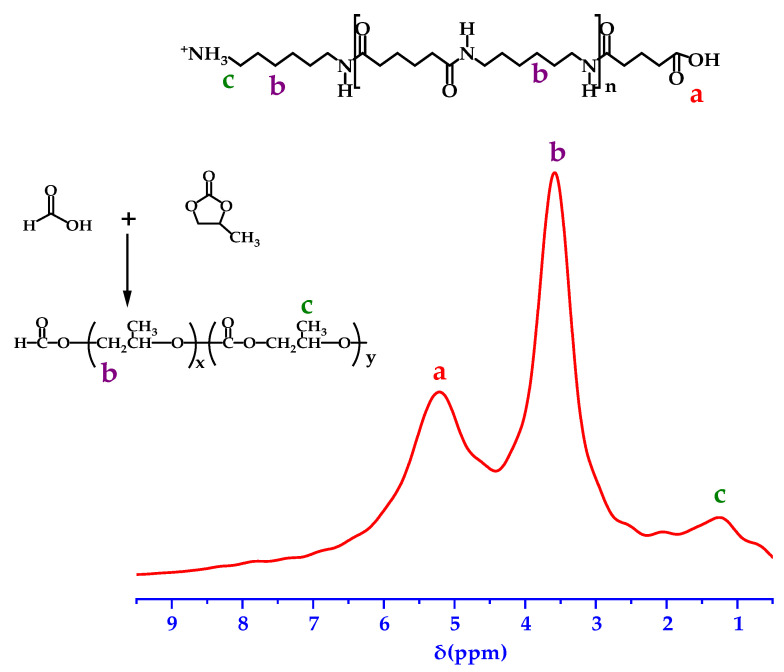
^1^H NMR of the P3 membrane.

**Figure 5 polymers-14-01706-f005:**
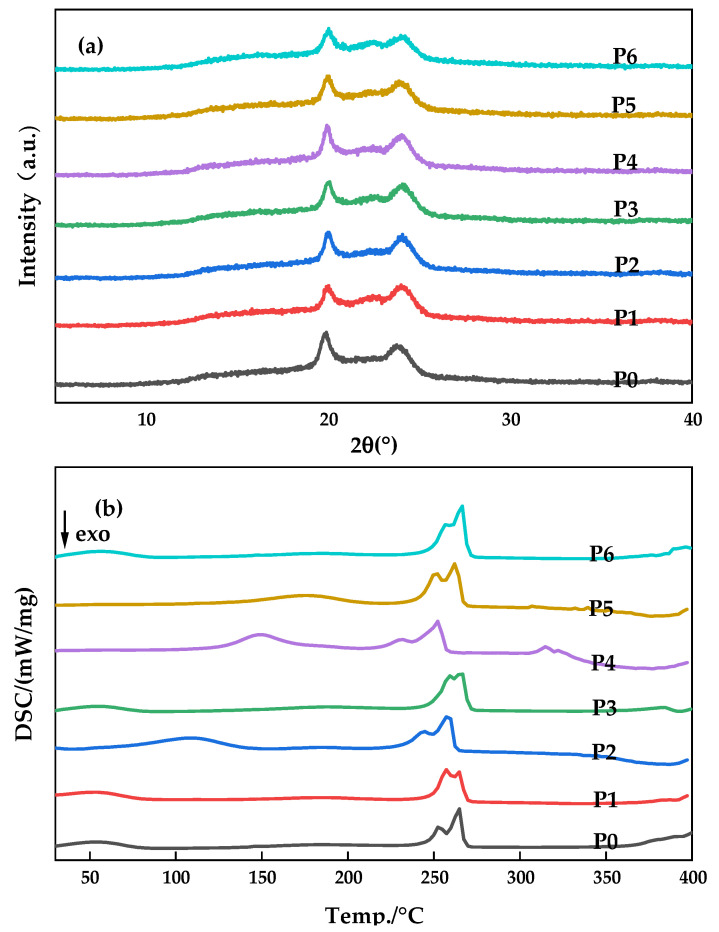
X-ray diffraction patterns (**a**) and DSC curves (**b**) of various PA66 membranes.

**Figure 6 polymers-14-01706-f006:**
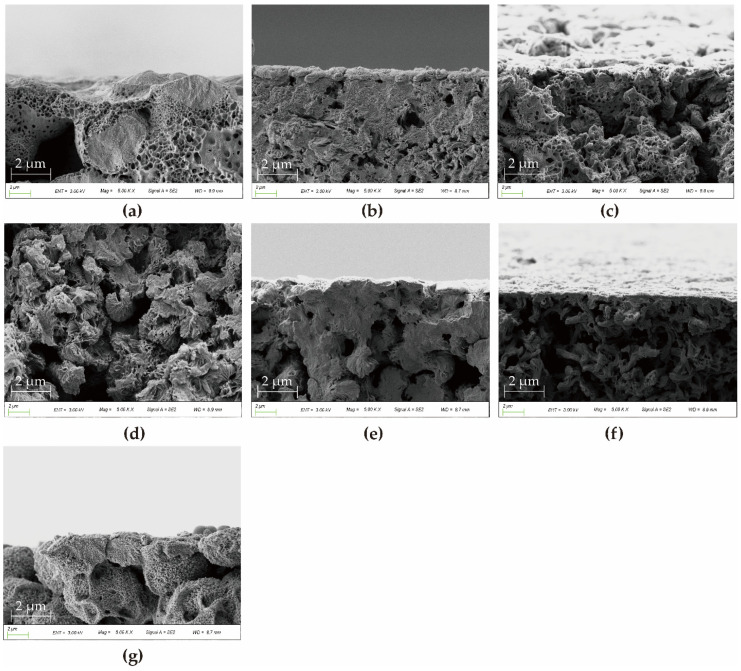
SEM images of cross-section of various PA66 membranes: (**a**) P0, (**b**) P1, (**c**) P2, (**d**) P3, (**e**) P4, (**f**) P5 and (**g**) P6.

**Figure 7 polymers-14-01706-f007:**
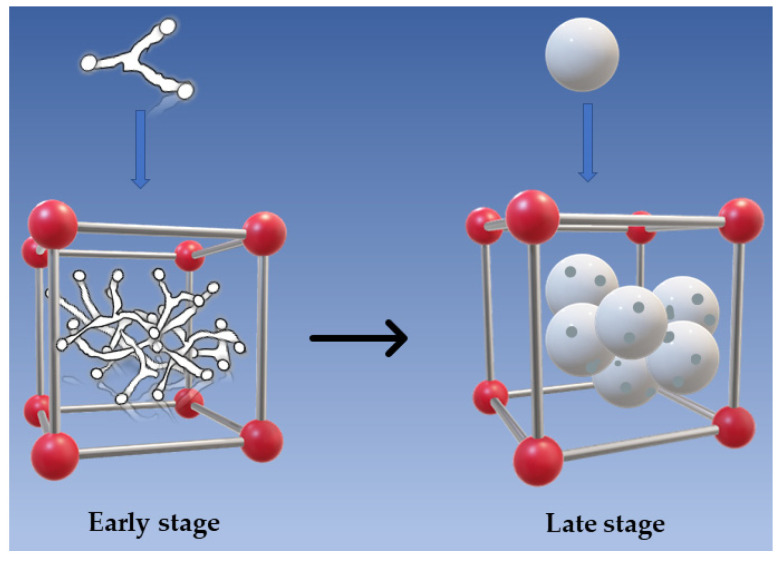
Crystal growth and evolution process.

**Figure 8 polymers-14-01706-f008:**
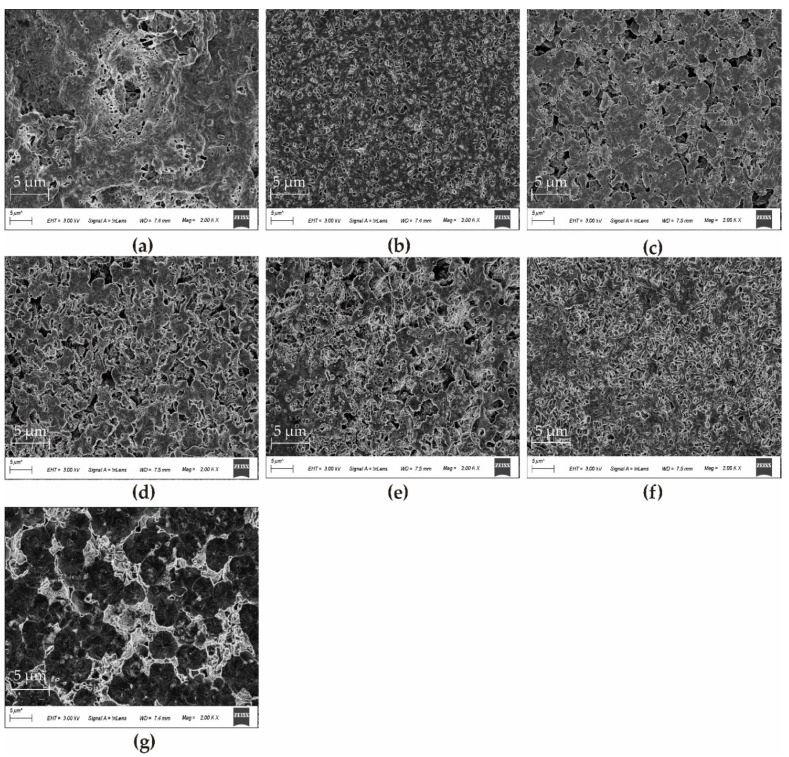
SEM images of upper surfaces of various PA66 membranes: (**a**) P0, (**b**) P1, (**c**) P2, (**d**) P3, € P4, (**f**) P5, and (**g**) P6.

**Figure 9 polymers-14-01706-f009:**
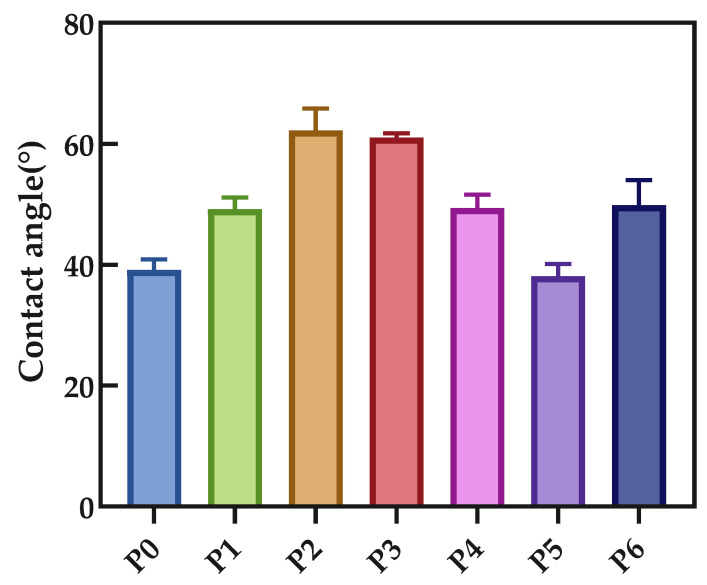
Contact angle of various PA66 membranes.

**Figure 10 polymers-14-01706-f010:**
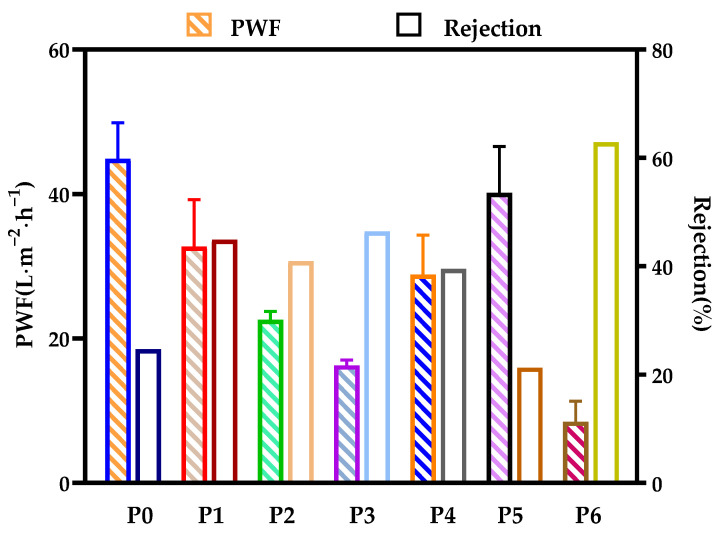
Pure water flux and BSA rejection of various PA66 membranes.

**Figure 11 polymers-14-01706-f011:**
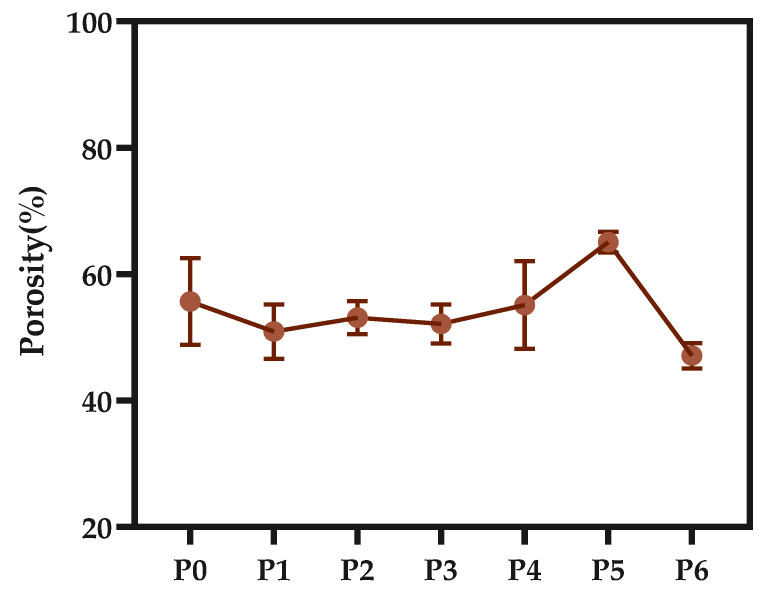
Porosity of various PA66 membranes.

**Figure 12 polymers-14-01706-f012:**
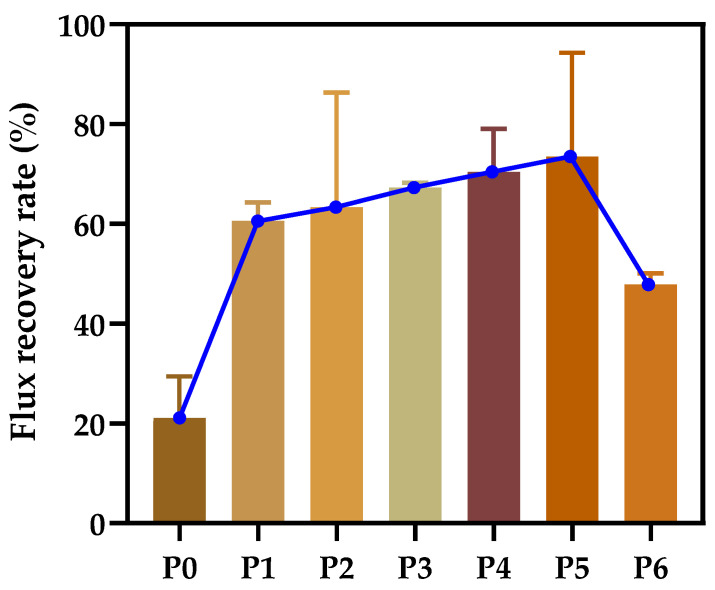
Flux recovery rate of various PA66 membranes.

**Figure 13 polymers-14-01706-f013:**
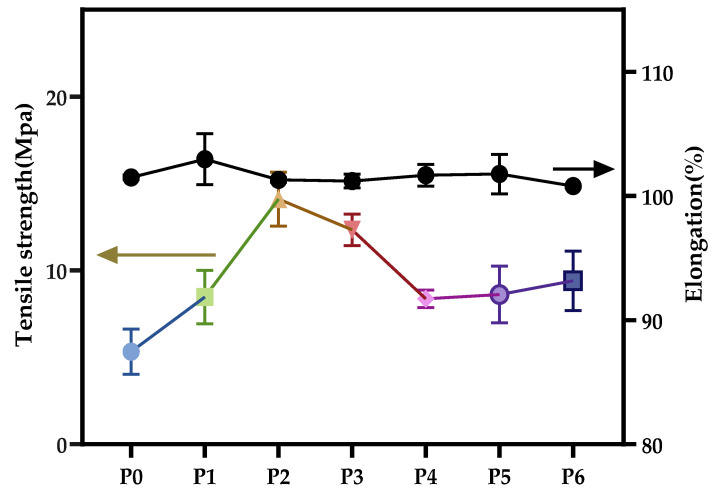
Tensile strength and elongation at break of various PA66 membranes.

**Table 1 polymers-14-01706-t001:** Casting solution composition and preparation conditions of different membranes.

Code		Content (wt.%)		Aging	Coagulation Bath
	PA66	FA	PC	Time (h)	Temperature (°C)
P0	20	80	0	2	15
P1	22	78	0	2	15
P2	22	73	5	2	15
P3	22	68	10	2	15
P4	22	63	15	2	15
P5	22	58	20	2	15
P6	24	76	0	2	15

**Table 2 polymers-14-01706-t002:** Elemental composition by atomic percent of different kinds of PA66 membranes.

Membrane	C (%)	N (%)	O (%)
P1	70.52	10.69	18.79
P3	73.75	11.60	14.65

## Data Availability

The data used to support the findings of this study are available from the corresponding author upon request.

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
