# Peer review of "Axial Crystal Growth Evolution and Crystallization Characteristics of Bi-Continuous Polyamide 66 Membranes Prepared via the Cold Non-Solvent-Induced Phase Separation Technique"

_polymers, 2022, doi:10.3390/polym14091706_

Round 1

Reviewer 1 Report

Wang et al. reported cold solvent-induced phase separation method for crystal growth of polyamide 66 microporous membranes. The membrane structure is systematically characterized by the XPS, FTIR, XRD, SEM etc. The study well written and systematically study and well organized in the manuscript. Therefore, I recommend this manuscript for acceptance in polymers after addressing the following issues given below.

  • First big flaw in this manuscript is two statements are completely in opposite meanings: in the title, Authors claiming bi-continuous polyamide 66 membranes prepared via the cold-non-solvent-induced phase separation technique; while in the abstract and preparation section they use solvent for membrane preparation (PC and farmic acid).

  • In characterization part at line 179, PC changes the degree of polymerization between copolymers: the statement is not clear enough: is there any polymerization process during solvation polyamide in PC and farmic acid?

  • The scale used in SEM images is not consistent, therefore not possible to develop comparison between different samples, Secondly, the scale is blurred and difficult to read.
  • Author provided only FTIR to confirm the composition of the membrane. The use of strong acid probably initiate the ring opening polymerization of PC, therefore, author must provide the NMR of the sample to further confirm the composition of the samples.

Reviewer 2 Report

This work shows quite interesting results on the synthesis, characterization and application of PA66, but this manuscript has to go through very substantial revision:

1) XPS. Figure 1. Your XPS fitting is absolutely confusing and wrong!!! :

a) C1s curve fitting shows only C-C contribution with some shift?What is it Fig 1b &c? Please provide a correct fit for C1s

b) Fig 1d and 1e prsnts unacceptable fit. You cant mix so many contribution in a one peak spectrum!!!! Please do not use contributions with so different FWHM! Please read some articles or address some help from XPS experts. 

c) The speculations about C-N shift is nonsense if one take into account my comments above

2) WCA results. Please indicate how many samples, droplets were taken? Please provide statistics.

3) In general all data (Figs8,9,10 11 and etc) must shows the standard error deviations.

Round 2

Reviewer 1 Report

The most part of the manuscript is revised well. The still the title and description in the experimental section is not consistent. The Authors adopted the method for the synthesis of membrane (just changed the materials and solvent) reported by the Cheng et al. (Effects of bath temperature on the morphology and performance of EVOH membranes prepared by the cold-solvent induced phase separation (CIPS) method, J. APPL. POLYM. SCI.2017, DOI: 10.1002/APP.44553). As author used PC and farmic acid as solvent, the title should be “Axial crystal growth evolution and crystallization characteristics of bi-continuous polyamide 66 membranes prepared via the cold-solvent induced phase separation technique”and accordingly change in the experimental section and other section of the manuscript  

Reviewer 2 Report

All addressed questions were answered and all suggested changes were provided in the revised manuscript.
